# Relating Physical Activity to Problematic Internet Use

**Killian Conyngham**[*]
Master of Data Science for Public Policy
Hertie School, Berlin, Germany
k.conyngham@students.hertie-school.org

**Jackson M Luckey**[*]
Master of Data Science for Public Policy
Hertie School, Berlin, Germany
jacksonmluckey@gmail.com

**Fabian Pawelczyk**[*]
Master of Data Science for Public Policy
Hertie School, Berlin, Germany
f.pawelczyk@yahoo.de

## Abstract

The focus of our project is to predict the risk of problematic internet usage (PIU) by minors, based on data of their physical activities. In particular, our goal is to build a model for entrance in the Kaggle competition run by the Child Mind Institute on this topic. Here, one key challenge lies in the high level of missing data—particularly concerning the target variable—and the actigraphy time-series component of the dataset. Our primary objective will be to design and train a deep neural network to effectively address these issues.

## 1  Background

Problematic Internet Use (PIU) has become a global phenomenon in recent years, with prevalence estimates ranging between 20% and 45% [7]. PIU is defined as excessive and compulsive internet use and is associated with increased symptoms of depression, anxiety, ADHD, and aggression [16, 7, 13], as well as reduced physical activity and life satisfaction [7, 3, 14, 2]. Children and adolescents are particularly vulnerable to PIU, given their developmental stage and the critical impact of early behaviors on long-term health and social outcomes [22, 12].

Measuring PIU in children and adolescents often relies on complex, professional evaluations, creating accessibility challenges due to cultural, linguistic, and resource barriers. In contrast, physical fitness metrics—such as activity levels—are widely available and accessible without clinical expertise. Using these physical indicators (i.e. data gathered with accelerometers) as proxies to identify PIU offers a practical alternative where direct assessments are not feasible. Therefore, predicting PIU from physical indicators helps to establish an early-warning system to inform health and education policies with evidence that can guide resource allocation efficiently.

## 2  Definitions

Given the challenge, our goal is to find the optimal prediction model for our target variable, the Severity Impairment Index (SII)—a standard measure of problematic internet use. The optimal model is defined as the one that maximizes the quadratic weighted kappa (QWK), and thereby the loss function we will focus on is:

$$\kappa = 1 - \frac{\sum_{i,j} w_{ij} O_{ij}}{\sum_{i,j} w_{ij} E_{ij}} \tag{1}$$

---

[*]These authors contributed equally to this work.

38th Conference on Neural Information Processing Systems (NeurIPS 2024).

where $O_{ij}$ is the observed ratings matrix, $E_{ij}$ is the expected ratings matrix, and $w_{ij}$ is the weight matrix defined as:

$$w_{ij} = \frac{(i-j)^2}{(N-1)^2} \tag{2}$$

with $N$ being the number of possible rating categories. The expected matrix $E_{ij}$ is calculated as:

$$E_{ij} = \frac{(\sum_k O_{ik})(\sum_k O_{kj})}{n} \tag{3}$$

where $n$ is the total number of rated items.

## 3    Related Work

Conducting a review of the top Kaggle submissions, we see that the most common approaches are tree-based ensemble learning approaches. Specifically, the top submission with public code uses an ensemble of LightGBM, XGBoost, and CatBoost through a Voting Regressor. This approach achieves a QWK of 0.494, third place overall. As the two submissions above it achieve only marginally better performance of 0.495 and 0.497, it is likely that they employ a similar approach. Almost all the top performing models use a similar architecture, motivating us to begin with a tree-based ensemble approach as a reference, which we can hopefully outperform with our deep neural network.

More generally, there are some key considerations which must be addressed for fitting any model on this dataset. The accelerometer data has significant missingness and is absent entirely for some participants. Furthermore, data for some participants does not differentiate between non-wear and sedentary activity. Previous research has used multiple imputation through chained equations (MICE) and other regression approaches, tree-based models, pooling models, and autoencoders to impute missing actigraphy data [1, 5, 11, 19, 20]. With sufficient data per participant (5+ days of continuous wear or 7+ days with 10+ hours of data each), imputing missing values using simple statistical methods (e.g., mean, median) has been shown to have non-catastrophic effects on results of other psychiatric research [5, 20]. Differentiating between non-wear and sedentary activity is challenging, but existing research using heuristics, logistic regression, and tree-based models has been successful, albeit on larger datasets [18, 19]. For children, separating out school days from non-school days is recommended [5]. The label is missing for a portion of the training data. Other researchers have had success applying pseudo-labelling and other semi-supervised methods to time series with partially labelled training data in related health domains[15, 4, 9].

## 4    Proposed Method

The Kaggle challenge provides the dataset. The training data consists of 3800 samples, each representing a 5 to 22 year-old. For each sample, we have a row of tabular data on health and demographic characteristics, as well as time series data from the accelerometer. The challenge does allow for the use of other open-source data as external sources, as the data is anonymised, however, matching may be a challenge, so we have not actively considered external datasets yet. We will use a semi-supervised learning (i.e., pseudo-labelling) approach to handle the missing labels[15, 9, 4] and try regression, tree-based, and deep learning methods to impute gaps in the actigraphy time series data[1, 5, 11, 19, 20, 18, 17].

We propose to fit a deep neural network as our main classifier for this problem, in line withe recent literature on the strong empirical results of deep learning applied to time series classification [10]. In particular, we would fit a Hybrid Neural network incorporating both the time series and tabular data, an approach shown to be effective with sensor data [21]. For the time series data, we plan on considering both Long Term Short Memory (LSTM) and Gated Recurrent Unit (GRU) architectures, with an emphasis on the GRU architecture due to its strong empirical performance on smaller datasets with short sequences[17, 8, 6].

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
