# OpenReview forum: "[Proposal-ML] Relating Physical Activity to Problematic Internet Use"
_tsinghua.edu.cn/THU/2024/Fall/AML — THU 2024 Fall AML Submission_

### Official Review · ~Guilherme_Félix_Diogo1 · 2024-11-06
**Clear explanation of the problem but more could be said on the models**

**Rating:** 8
**Confidence:** 4

**Review:**

The proposal discusses a relevant and socially significant issue, which is predicting problematic internet usage (PIU) of underage individuals driven by physical activity data. The project is relevant and supported by a great deal of existing literature, especially with regards to the use of physical activity data as a proxy for behavioral health measures. The employment of deep learning approaches for this objective seems to be justified owing to the characteristics of the data. A little bit more could be said on how better performing neural network systems built deep learning will deep learn tree models into extinction.

---

### Official Review · ~Lily_Sheng1 · 2024-11-07
**Review of Submission 51**

**Rating:** 8
**Confidence:** 4

**Review:**

The submission addresses the issue of Problematic Internet Use (PIU) and using physical indicators to predict PIU in children and adolescents. The approach employs a semi-supervised learning approach with a Hybrid Neural Network, combining LSTM or GRU architectures, to handle missing data and integrate accelerometer time series with health and demographic data for PIU prediction.

Pros:
1. The use of imputation and semi-supervised learning addresses missing data challenges.
2. The topic aligns well with current research trends, as it combines machine learning techniques and physical activity data to address behavioral health issues like PIU.

Cons:
1. There could be more details on what the evaluation metric would be for different choices of architectures.

---

### Official Review · ~Jinsong_Xiao1 · 2024-11-09
**review for proposal 51**

**Rating:** 8
**Confidence:** 4

**Review:**

This proposal investigates the link between physical activity data and the risk of problematic internet use (PIU) among minors, with a focus on designing a deep neural network for PIU prediction.

Pros:

- Clear Objectives: The authors outline the problem effectively, with a well-defined target (Severity Impairment Index) and evaluation metric (Quadratic Weighted Kappa).

- The use of a hybrid neural network to integrate time-series and tabular data is logical and supported by recent literature.

Suggestions: A more detailed introduction of the evaluation scheme and the application of the new model

---

### Official Review · ~André_Moreira_Leal_Leonor1 · 2024-11-09
**Proposal 51 review**

**Rating:** 8
**Confidence:** 4

**Review:**

The important, very timely issue that this proposal tries to address is the prediction of problematic Internet use among minors, using physical activity data as a proxy for psychological risk factors. In so doing, this approach considers not only invasive assessments but also the data that are widespread, hence underpinning the potential societal impact of the project. Deep learning methods apply best to such complex data, with missing values inherent in accelerometer readings; this justifies the methods chosen in reaching the goals of the project.

The application of hybrid neural networks with semi-supervised learning methods is a well-thought-out approach in improving prediction accuracy by using techniques such as pseudo-labeling. While it is evident why deep learning is applicable, the proposal could go further in explaining how the chosen model architecture might outperform traditional tree-based ensemble methods, which may highlight special advantages of deep learning in dealing with time series and missing data in behavioral prediction.

---

### Official Review · ~Zhijie_shen3 · 2024-11-10
**51 Review**

**Rating:** 8
**Confidence:** 3

**Review:**

### Peer Review
**Summary**
The project has strong potential with a well-thought-out approach.

**Pros**:
1. The choice of LSTM and GRU for time-series analysis is appropriate given their proven effectiveness in handling sequential data. The proposal shows a good understanding of leveraging deep learning for complex patterns.
2. The use of semi-supervised learning with pseudo-labeling to handle missing data is innovative and can enhance the model's robustness. This approach shows creativity in addressing data challenges.
3. Combining accelerometer time-series data with tabular demographic data is a strong approach. This comprehensive integration could improve the accuracy of predicting problematic internet use.

**Suggestiong**:
1. The proposal lacks specific details on how the deep learning models will be optimized, such as hyperparameter tuning or strategies to prevent overfitting. Consider using techniques like cross-validation and dropout.
2.  While pseudo-labeling is a creative solution, it may not be sufficient for large data gaps. Exploring additional imputation methods, such as RNN-based imputation or interpolation, could improve data quality.
3. While using QWK is a good choice, the proposal would benefit from including additional metrics like F1 score, precision, and recall to provide a more comprehensive evaluation of model performance.

---

### Official Review · ~Yang_Ouyang2 · 2024-11-11
**Well-researched approach, but would benefit from clearer model justification and more discussion of evaluation methods.**

**Rating:** 10
**Confidence:** 4

**Review:**

Strengths
Relevant Problem Focus: Children and health is a big concern of current age.
Thorough Literature Review
Clear Objective and Methodology: The authors have identified the target variable, Severity Impairment Index, and the primary goal of maximizing the quadratic weighted kappa metric.

Weaknesses
Lacks Justification for for choosing specific models.
Limited Discussion on evaluation. Will there be cross evaluation?

The proposal presents a well-researched approach, but would benefit from clearer model justification and more discussion of evaluation methods.

---

### Official Review · ~Cheng_Gao2 · 2024-11-12
**Review for Relating Physical Activity to Problematic Internet Use**

**Rating:** 8
**Confidence:** 4

**Review:**

Strengths:

- The issue highlighted is a highly relevant topic is practical and innovative.
- The methodological approach is well thought out, with a clear plan to use deep learning models and a hybrid neural network to integrate time series and tabular data.
- The use of semi-supervised learning (pseudo-labelling) to handle missing labels and techniques like GRU for time series is promising and aligns with current best practices.
- The proposal is thorough in its consideration of data limitations and shows an understanding of the challenges in data imputation and model design.

Weaknesses:

- The proposal could benefit from a **clearer task definition**. For instance, specifying the exact format and type of model input would enhance the reader's understanding of this task.
- There could be more discussion on potential **evaluation strategies** and error analysis methods to ensure model robustness, particularly for edge cases in the data.

---

### Official Review · ~Zhuofan_Sun1 · 2024-11-12

**Rating:** 10
**Confidence:** 5

**Review:**

This paper proposes a deep learning approach to predict Problematic Internet Use (PIU) in minors based on their physical activity data. The authors aim to participate in a Kaggle competition on this topic and focus on developing a model that maximizes the quadratic weighted kappa (QWK) metric. The paper presents a clear background on PIU, defines the problem, and outlines related work.
Strengths:
Relevance and Importance: The study addresses a critical issue affecting a growing number of adolescents, with potential negative consequences for their mental and physical health.
Clear Objective: The paper clearly defines the goal of predicting PIU severity using physical activity data, measured through accelerometers.
Methodological Approach: The authors propose a comprehensive approach involving semi-supervised learning for handling missing labels and a deep neural network incorporating both time series and tabular data.
Technical Considerations: The paper acknowledges and addresses key challenges, such as missing data in accelerometer data and the need to differentiate between non-wear and sedentary activity.
References: The paper provides a solid foundation with relevant literature on PIU, time series classification, and deep learning techniques.
Overall, this paper presents a promising approach to predicting PIU using physical activity data. Addressing the aforementioned weaknesses would strengthen the paper and provide a more comprehensive evaluation of the proposed model.

---

### Official Review · ~Isak_Tønnesen1 · 2024-11-12
**Review: [Proposal-ML] Relating Physical Activity to Problematic Internet Use**

**Rating:** 10
**Confidence:** 4

**Review:**

This proposal presents a highly relevant and well-structured approach to predicting problematic internet use (PIU) in minors using physical activity data. The methodology is particularly strong, combining state-of-the-art deep learning techniques (hybrid neural networks with LSTM/GRU) with innovative solutions for handling missing data through semi-supervised learning. The authors demonstrate good understanding of both the technical challenges and societal importance, supported by a promising literature review. The clear mathematical formulation of the problem, detailed evaluation metrics (QWK), and thoughtful consideration of data limitations make this a interesting proposal. The choice to integrate both time-series actigraphy data and tabular demographic information shows good understanding of the problem space. While minor clarification on model architecture specifics would be beneficial, the overall approach is thorough and promising for advancing both PIU detection and general behavioral health monitoring.

---

### Official Review · ~Han-Xi_Zhu1 · 2024-11-12
**Review for Relating Physical Activity to Problematic Internet Use**

**Rating:** 9
**Confidence:** 4

**Review:**

The proposal focuses on a concerned social problem, problematic internet usage (PIU). The authors try to design and train a deep neural network to predict the risk of PIU.

## Strengths


1. The author propose a hybrid neural network to integrate time series and tabular data, which is solid and practical.
2. The problem the author try to address is a globally concerned issue. It has significant implications for public health and social outcomes, particularly among young populations.


## Weaknesses
1. While QWK is mentioned as the primary metric, a more comprehensive evaluation strategy, including additional metrics and validation techniques, would strengthen the project.

---

### Official Review · ~Wuqian1 · 2024-11-12
**Review of "Relating Physical Activity to Problematic Internet Use"**

**Rating:** 8
**Confidence:** 4

**Review:**

The proposal "Relating Physical Activity to Problematic Internet Use" is of high quality. It addresses a significant public health concern, the risk of problematic internet usage (PIU) among minors, through the lens of physical activity data. The project's approach to building a predictive model using deep neural networks is technically sound and aligns with current trends in data science and machine learning.
Pros
   1.Innovative Approach: Utilizes a deep neural network to predict PIU, offering a potentially more accurate and nuanced prediction model.
   2.Technical Rigor: The proposed method includes sophisticated techniques like LSTM and GRU architectures for time series data.
Cons
   1.Complexity of Data: The project involves complex data, including missing data and the need for imputation, which could introduce errors or biases.
   2.Dependence on Kaggle Data: The project relies heavily on the Kaggle dataset provided by the Child Mind Institute, which may limit the generalizability of the findings.

---

### Official Review · ~liyingxin1 · 2024-11-12
**Should add more details about the method of handling time series data**

**Rating:** 9
**Confidence:** 4

**Review:**

Very meaningful topic. Persuasive method and detailed related works.

I think time series data has its difficulty to prepare. So it is suggested to elaborate on the methods of data handling, especially in dealing with missing data and time series data, to ensure the reproducibility and reliability of the methods.